# Do Increased Tax Base and Reductions in the Underground Economy Compensate for Lost Tax Revenue Following a Tax Reduction Policy? Evidence from Italy 1982 to 2006

**Renzo Orsi [1] and Knut Lehre Seip [2],***

1   Department of Economics, University of Bologna, 40126 Bologna, Italy; renzo.orsi@unibo.it
2   Department of Technology, Arts and Design, Oslo Metropolitan University, P.O. Box 4, St. Olavs Plass, Pilestredet Park 33, 0176 Oslo, Norway
*   Correspondence: knut.lehre.seip@oslomet.no

**Abstract:** We here examine the frequent claim that an increase in the tax base and a decrease in tax evasion will compensate for a loss in tax revenues caused by a lower tax level. Using a unique data set for the estimated underground economy in Italy from 1982 to 2006, we found that a loss in tax revenues equivalent to 1% of the GDP would be partly compensated by an increase in GDP of 0.55%. The compensation would come from 0.31% of the GDP increase and from 0.24% of the reductions in the underground economy. These results apply to an economy with a high tax level (>32%) and a high underground economy (≥25%). Applying a high-resolution lead–lag method to the data, we ensured that tax changes were leading the GDP and, thus, a potential cause for changes in the GDP.

**Keywords:** tax policy; GDP; underground economy; tax evasion; self-financing

**JEL Classification:** O17; C63; E52; H26

## 1. Introduction

Revenue loss from tax reductions is often claimed to be compensated by increases in the gross domestic product (GDP) and decreases in the underground economy (UGE) by both the popular press (Krugman 2019) and politicians (Muller et al. 2016). However, the literature is not conclusive on the effects of tax reductions (Mountford and Uhlig 2009; Trabandt and Uhlig 2011; Arin et al. 2013; Seip 2019), or on the effects of tax evasion (Romer and Romer 2010; Muehlbacher et al. 2011). One reason is that a change in tax policy will have an effect over interannual time horizons, which over longer time periods may diminish or merge with other effects or changes in tax policies.

Here, we examine if a decrease in taxes, increases the GDP and decreases the underground economy (UGE), and if the two effects are sufficient to compensate for the loss in tax revenues at the lower tax level. The objective of the study is formulated in three hypotheses below and supported by rationales for the reasonability of the hypotheses. An examination of the results in the context of previous studies on taxation, the GDP, and the UGE is given in the discussion section.

We use a high-resolution lead–lag (HRLL) technique that allows us to identify lead–lag (LL) relations over very short time windows (n = 3 consecutive and synoptic observations, with n = 9, significance levels that can be identified, Seip and Grøn (2018)). The short time windows allow us to see how a leading role for one variable may change to a lagging role after a short period of ≈10 time steps. Alternative LL methods require much longer time series, typically n = 30–40. In most cases, they require that the series are stationary and Gaussian, e.g., for cross-correlation techniques and non-linear state space reconstructions (Sugihara et al. 2012; Kestin et al. 1998).

In the present study, all series were first linearly detrended so that the focus was on the decadal changes in the variable. In other studies, the first difference is used to detrend the relevant economic time series. However, the first difference may shift peak and trough patterns relative to their dates in the raw series as well as introduce more high-frequency noise. Second, we identified time windows during the period from 1982 to 2006 when changes in tax policies led to changes in the GDP and the UGE, and thus enhanced the probability that the tax policies were the cause for changes in the GDP and the UGE. To our knowledge, there is only one other study, Seip (2019), that identified time windows where tax changes led to changes in the GDP. However, that study dealt with US data and did not address issues with tax evasion.

Our study is restricted to the period from 1982 to 2006 because we had reliable estimates from two independent sources for the amount of tax evasion during this period. Orsi et al. (2014) used DSGE modeling, and Chiarini et al. (2013) used estimates from value-added tax reports.

We found that decreasing total taxes (TT) increased the GDP and decreasing personal taxes (PT) decreased the UGE. Overall, based on the Italian economy during the period from 1982 to 2006, the loss in tax revenue from decreasing taxes was partly compensated by approximately one-half of the losses from increasing the GDP and decreasing the UGE. The contribution from the GDP increase was the largest (56%). Our findings are generic and can be applied to economies with a high tax burden greater than about 32%, and a sizeable UGE above about 25% of the economy. Thus, the equations we arrive at for the relations between tax burden, the GDP, and the UGE could contribute to estimating the effects of changing taxes.

*Hypotheses*

We suggest three hypotheses. First hypothesis, **H1**: When total taxes decrease, the GDP will increase over an interannual time scale. The rationale is that when taxes decrease, the private sector will use the surplus gain to invest and increase effectiveness. Second hypothesis, **H2**: When personal taxes increase, the UGE will increase, because, in the tradeoff between higher personal gain and the chance of being caught, tax evasion and personal gain will be favored. Third hypothesis, **H3**: The loss in tax revenues (TR) from a reasonable ($\approx$1–5%) reduction in tax rates will be compensated by the gain in taxes from a higher GDP and a lower UGE.

*GDP and UGE in Italy*. It is reasonable to believe that the effect of a tax policy depends on the tax level (Trabandt and Uhlig 2011) and on the mix of taxes. Our study uses data on taxation, the GDP, and the UGE in Italy from 1982 to 2006 (2006 being the last year where UGE data were calculated (Orsi et al. 2014)). Total taxes varied between 27% and 36% of the GDP and personal taxes rose from 25% in 1982 to 46% in 2006. The UGE ranged between 7% and 26% of the total economy. Thus, the Italian economy lends itself to the study of the effects of tax levels on the economy. However, the effects of tax policy may depend on the ability of an economy to harvest the benefits of a tax cut (Bloom et al. 2013) and on the tax morale of the society in which it is implemented (Barone and Mocetti 2011; Alm 2012; Moro-Egido and Solano-García 2020).

The rest of the paper is organized as follows. In Section 2, we present the data. In Section 3, we give an outline of the method, particularly the high-resolution lead–lag (LL) method. In Section 4 we show the results, and in Section 5 we discuss the results and outline policy implications. In Section 6, we conclude.

## 2. Data

Our target variables are the GDP and the UGE during the period from 1982 to 2006 in Italy. We use (i) TT as a candidate causal variable for changes in the GDP and (ii) PT as a candidate causal variable for changes in the UGE. We also examine (iii) tax evasion control efforts. The UGE refers to activities that are productive and legal but concealed from public authorities to evade being taxed or to avoid regulations. Tax evasion refers

to the part of the UGE that is concealed to avoid taxes on income, value-added taxes, or other taxes (Dell'Anno and Davidescu (2019) citing OECD (2002)). Both TT and PT were supplied as percentages of the GDP by Orsi et al. (2014) but were recalculated to monetary units (Euro) for some purposes. Tax revenues are the government's income from tax level (%) and changes in the tax base (number of people and firms paying taxes).

In a recent study, Orsi et al. (2014), taking a cue from the model of Allingham and Sandmo (1972), have proposed a dynamic stochastic general equilibrium model, DSGE, that allows for an estimate of the underground economy in Italy for the period 1982 to 2006. Personal, corporate, and social taxes were included as variables in the DSGE model for the UGE. The Italian Institute of Statistics (ISTAT 2007) provided data for consumption, investment, wages, and fiscal revenues. The UGE is expressed as percentages of GDP. A second source for an estimate of tax evasion is given in Chiarini et al. (2013). The authors constructed a quarterly time series of tax evasion for the period 1980:1–2006:4 using the annual value-added taxes, VAT, estimated by the Italian Revenue Agency. The two estimates measure different aspects of tax losses and will be compared in the discussion section. The proportion of fiscal controls (Cp) is given by the ratio between the number of inspections and the number of companies susceptible to inspection on an annual basis. The time series were provided by the Agenzia delle Entrate (the Italian Revenue Agency) and have been made quarterly by Orsi et al. (2014).

There are several theories for factors that will change UGE, e.g., in the seminal study by Sandmo (2005) and, recently in relation to the COVID-19 pandemic, Williams and Kayaoglu (2020). Based on suggestions by Williams and Kayaoglu (2020, p. 83) on the impacts of the level of GDP, we add an examination of the relation between changes in GDP, $\Delta$GDP = GDP$_{t+4}$ − GDP$_t$ and UGE.

*Tax policy events*. In the period after 1992, tax amnesties were introduced to reduce the number of pending lawsuits dealing with tax evasion. The most significant tax amnesty was in 1992, followed by a similar provision in 1993. In 2001, undeclared workers could voluntarily enter the formal workforce (Williams and Kayaoglu 2020). Later, from 2003 to 2004, a wide range of amnesty measures was offered to taxpayers to close tax disputes pending with the tax authorities. A survey of tax policy events in Italy is given in Chiarini et al. (2013, p. 279), and an assessment of the efficiency of fiscal control is given by Lisi and Pugno (2015, p. 358).

The Orsi et al. (2014) data are shown centered and normalized to unit standard deviation in Figure 1a. (See Section 3.1 on normalization below). The UGE time series by Orsi et al. (2014) and tax evasion series by Chiarini et al. (2013) are compared in Figure 1c. The figure shows the linear trend and the cyclic component of their UGE estimates. Figure 1d shows the time series for $\Delta$GDP and UGE.

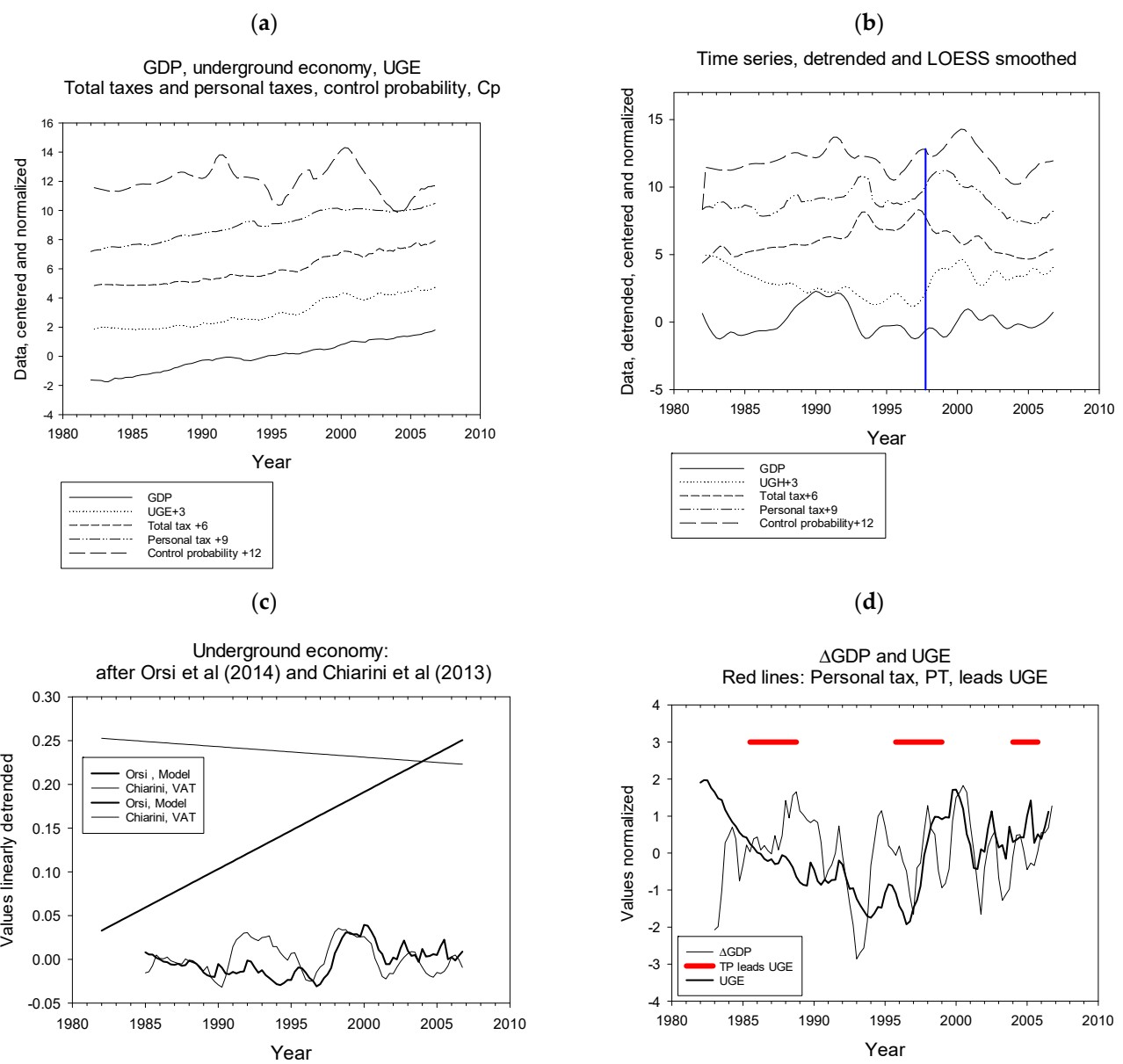

**Figure 1.** Time series. (**a**) Raw time series normalized to unit standard deviation and each series shifted vertically in steps of three units. (**b**) Series detrended and normalized. Drop-line(the blue line) shows the date 1997:4 when major tax shocks occurred in both personal and corporate taxes. (**c**) Comparison of two time series for UGE after Orsi et al. (2014) and tax evasion after Chiarini et al. (2013). The series are linearly detrended (trends: upper two straight lines). The residuals (lower two curves) are centered and normalized. (**d**) GDP first difference $\Delta GDP = GDP_{t+4} - GDP_t$ and UGE linearly detrended. Red horizontal lines show periods where PT leads UGE. GDP = gross domestic product, UGE = underground economy, TT = total taxes, PT = personal taxes.

## 3. Methodology

We first outline how we pretreated the data and thereafter our method for determining tax shocks. Lastly, we briefly present the high-resolution LL method for calculating lead–lag relations.

### 3.1. Detrending, Smoothing, and Normalizing

We first linearly detrended the Orsi et al. (2014) data and our analysis was based on these data. By centering and normalizing the data, we ensured that the choice of units did not impact the results when least-squares methods were applied to the data.

Chiarini et al. (2013) applied their analysis to the logarithm of the data and smoothed the data with the Hodrick and Prescott (1997) filter. The form of the noise on the observed data was not known (uniform or Gaussian, additive or multiplicative). Since the DSGE modelling studies included added stochastic elements, we slightly smoothed both the observed and the modelled series. For smoothing, we used the 2D LOESS smoothing algorithm SigmaPlot©. The algorithm is a locally weighted polynomial smoothing function. We used its parameter (f) to define local domains (f is the percentage of the full series) and a second-order polynomial function, (p) = 2, to interpolate. Since we always use p = 2, we used the nomenclature LOESS(f) to show the LOESS smoothing used. However, apart from LOESS smoothing, no parameters were included in the algorithm that defined leading or lagging relations between paired series. The series are shown as linearly detrended, slightly LOESS(0.1)-smoothed and normalized series in Figure 1b. The dropline shows the date 1997:4, when several adjustments were made to the tax regime.

### 3.2. Shocks

We estimated tax shocks by taking the first derivative of corporate, personal, and social security tax rates. By normalizing the rates to unit standard deviation and making histograms for the normalized rates, we identified tax shocks by comparing their distribution to a fitted Gaussian curve. Outliers were identified as rates at the tails of the distribution. Rate changes $\leq 3$ and $\geq 3$ for tax changes were used to find the dates where changes had been substantial. The procedure was similar for control probability. We also examined if there were shocks in UGE.

### 3.3. Lead-Lag (LL) Relations

The high-resolution LL method is relatively novel (Seip and McNown 2007), but has been applied in several contexts, e.g., paleontology (Seip et al. 2018) and economics (Seip et al. 2019). We used two sine functions with a common cycle period ($\lambda$) as an example: (i) The sine series that peaks less than $1/2\lambda$ before the other is defined as a leading series. However, the leading property applies to all parts of the series. (ii) One series is either leading or lagging the other series, and if the leading series is inverted, it becomes a lagging series. This latter property is relevant for the units applied to measure an economic parameter. For example, a trough in the central bank's interest rate is hypothesized to cause a peak in GDP; thus, it is a peak in the inverted rate, the interest rate reduction, which is assumed to peak before GDP. (iii) LL relations are calculated for three synoptic observations in the pared series and therefore the series do not have to be stationary.

The method is based on the dual representation of paired cyclic series, x(t) and y(t), as time series and as phase plots, with x(t) on the *x*-axis and y(t) on the *y*-axis. Recently, a similar method based on the dual representations and wavelet analysis has been described by Krüger (2021). An intuitive presentation of the first part of the method where the paired time series x(t) and y(t) are depicted in phase portraits is related to the standard Lissajous curves in electrical engineering. A visual diagram is provided on Wikipedia (2015). The second part, where we calculate rotational angles for trajectories in phase plots is related to the standard calculation of magnetic fields around an electric wire (Wikipedia 2023). The method is simple, implemented in one Excel sheet, and is available from the authors.

#### 3.3.1. Explaining LL Relations

The description closely follows the description given in Seip (2019). To illustrate the method, we use two sine functions. One is a pure sine function with the cycle period $\lambda$. The other is a sine function that is phase-shifted, (PS), with $+1/8\lambda$. In addition, we added a small random component to the last sine function to make the example a little more realistic.

The two sine functions are shown in a dual representation, as time series along a time axis, Figure 2a, and in a phase plot with one series depicted on the *x*-axis and the other series depicted on the *y*-axis. In this example, the trajectories for the x(t), y(t) pairs

will rotate in a clockwise direction, as in Figure 2b. Pairs of ideal cyclic time series that are centered and normalized to unit standard deviation will show an elliptic form with center in the origin. With the major axis along the x = *y*-axis and with a phase shift of less than ¼ of a cycle period, the two series are pro-cyclic. With the major axis along the x = −*y*-axis and with phase shift in the range ¼λ to ½λ, the two series are counter-cyclic. If the trajectory rotates positively (counterclockwise per definition) then the *x*-axis variable leads the *y*-axis variable.

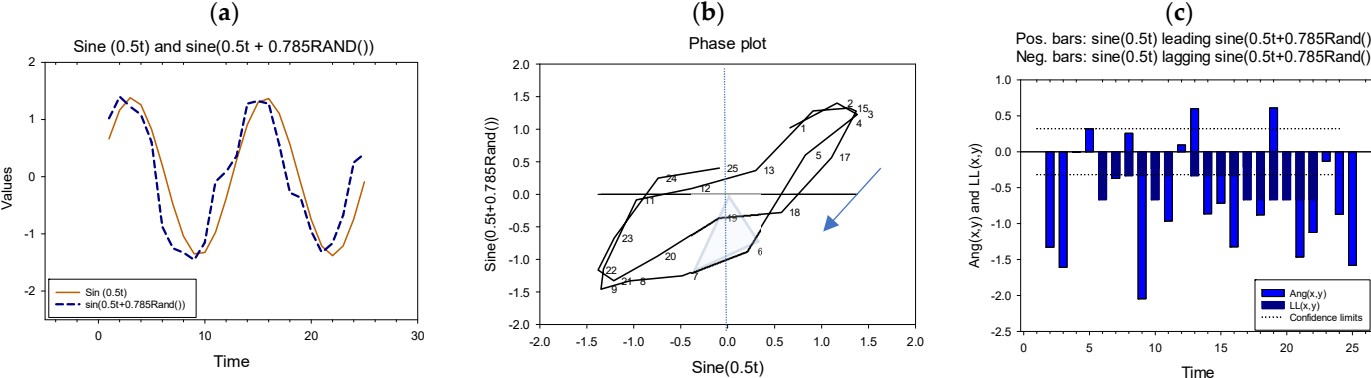

**Figure 2.** Example: Calculating leading–lagging (LL) relations and LL–strength. (**a**) Two sine functions: the smooth curve is a simple sine function, sin (0.5t), and the dashed curve has the form sin (0.5t + ϕ × RAND()) where ϕ = +0.785. (**b**) In a phase plot with sin (0.5t) on the *x*-axis and sin(0.5t + ϕ × RAND()) on the *y*-axis, the time series rotates clockwise; θ is the angle between two consecutive trajectories. The wedge suggests the angle between the origin and lines for observations 6 and 7. (**c**) Angles between successive trajectories (light blue bars) and LL strength (dark blue bars). Dashed lines suggest confidence limits for persistent rotation in the phase plot and persistent leading or lagging relations in the time series plot. Figure adapted from Seip and Zhang (2022).

We quantify the rotational patterns in phase plot using the function:

$$\theta = sign(\overline{v}_1 \times \overline{v}_2) \cdot A\cos\left(\frac{\overline{v}_1 \cdot \overline{v}_2}{|\overline{v}_1| \cdot |\overline{v}_2|}\right) \tag{1}$$

where $\overline{v}_1$ and $\overline{v}_2$ are two vectors formed by two sequential trajectories between three sequential points in the phase plots.

From these angles we identify a lead–lag (LL) strength. It is defined as a function of the number of positive angles, $N_{pos}$, minus the number of negative angles, $N_{neg}$, divided by the sum of the absolute values of both positive and negative angles over a certain time span.

$$LL = (N_{pos} - N_{neg})/(N_{pos} + N_{neg}) \tag{2}$$

The variable LL range between −1 (y-variable leads x-variable) to +1 (x-variable leads y-variable). Within a 95% confidence interval, (CI), −CI < LL < +CI, CI, there is no LL relationship, and no significant persistent cyclic variations in the time series plots. Here, we use $N_{pos} + N_{neg} = 9$.

### 3.3.2. Cycle Periods (CP)

One cycle period is λ ≈ 12.56 for the sine functions, as can be seen in Figure 2a. For a pair of perfect sine functions with a constant phase shift, a phase plot would show that the trajectory 1 to 13 would almost form a closed ellipse, and we can calculate and sum the angles between the origin and the points 1–13 to obtain a total angle of about 2π. Since the points 1–13 represent the time steps for the time series, ≈13 time steps also show the cycle period, λ, between two peaks in the time series. Thus, to obtain the cycle periods, λ, we

counted the number of time steps required for closure of the trajectories in phase space. This method is hereafter called the cumulative angle method.

### 3.3.3. Phase Shifts (PS) between Paired Series

The time series in Figure 2a are close, suggesting that the correlation coefficient between them may express the distance between them; that is, the phase shift PS. A PS of $\frac{1}{2}$ the common cycle period, λ, would suggest that the series are counter-cyclic and the regression coefficient, r = −1. A PS of zero would show that the series moves in perfect concert and are pro-cyclic. However, since the series show cycles with different lengths of the cycle periods, we must know the cycle period, λ, in advance to calculate the PS between them. PS is therefore estimated from the correlation coefficient, (r), for sequences of n = 5 observations, PS (5). Five observations are too short compared to the anticipated cycle periods. Thus, the PS will partly overestimate and partly underestimate the PS as the time window moves around the elliptic representation of the cyclic series in phase space. We calculated PS both with the moving average CP and with the average CP for the full time series. An expression for the phase shift between two cyclic series can be approximated by Equation (3):

$$PS \approx \lambda/2 \times (\pi/2 - \text{Arcsine}\,(r)) \tag{3}$$

For perfect sine functions, the phase shift is a function of the ratio between the major and the minor axes in the ellipse, Figure 2b.

### 3.3.4. Uncertainty Estimates

Using the Monte Carlo technique, we identified the 95% confidence interval (CI) as LL < −0.32 or LL > +0.32. The relationships are significant for these values if n > 9 (Seip and McNown 2015). The running average of LL was thereafter calculated over 9 successive observations. The number 9 is a tradeoff between the objective of calculating a CI and the objective of preserving a high-resolution LL measure. If the data is smoothed, the probability of detecting LL values with the same sign increases. Thus, the real CI will be larger.

### 3.3.5. Calculating GDP, UGE, and Tax Regressions

To calculate regressions between GDP, UGE, and tax variables, we use the original, but detrended variables. The regression equations will have the form:

$$\text{GDP}\,(10^9\ \text{Euro}) = \beta_1\ \text{TT}\,(\%) + \gamma_1 \tag{4}$$

$$\text{UGE}\,(10^9\ \text{Euro}) = \beta_2\ \text{PT}\,(\%) + \gamma_2 \tag{5}$$

GDP, UGE, TT, and PT are as defined before, and $\beta_1$ and $\beta_2$ are the regression coefficients for the GDP regression and the UGE regressions, respectively. The parameters $\gamma_1$ and $\gamma_2$ are the constants.

### 3.4. Self-Financing

Self-financing would mean that the loss in tax revenues caused by decreasing the tax level would be compensated by an increase in tax base (there are more taxpayers, and the taxpayers pay more tax because their income increases due to the tax cut) and because of a reduction in the underground economy. There might be some double counting depending upon the type of UGE that is reduced. We assume that all revenues collected above the revenues obtained before tax reduction are allocated to development with direct impact on GDP.

**Table 1.** Tax and economy shocks.

| Personal Tax | Corporate Tax | Social Security Tax | Tax Control Probability | UGE Volatility |
|---|---|---|---|---|
| 1989:4 (+3.2) **TT1** | 1991:4 (−2.8) **TT2** | 1989:4 (2.9) | 1990:4 (1.56) | |
| 1992:4 (+3.6) **TT2** | 1992:4 (+3.3) **TT2** | | 1991:4 (−1.38) **TT2** | |
| 1993:4 (−3.7) **TT2** | | | | |
| 1997:4 (+3.8) **PT2** | 1997:4 (+3.2) | 1997:4 (−7.5) | 1997:4 (−1.38) | |
| | 1999:4 (+2.5) | | | |
| | 2000:4 (+4.1) **TT3** | | | |
| | 2001:4 (−2.5) **TT3** | | | |
| | 2005:4 (2.4) | | | 2005:4 (−3.5) |

TTn = included in Total tax/GDP window number n; PTn = included in personal tax/UGE window number n.

The tax revenue (TR) with 1% tax reduction would be:

$$TR_{t+1} \approx TR_t \times 0.99 + (\beta_1 \times 1 \times \gamma_1) + TR_t/GDP_t + (\beta_2 \times 1 \times \gamma_2) \tag{6}$$

where $TR_t/GDP_t$ is the fraction of $GDP_t$ that is recovered as tax revenues. To find the change in GDP with changing TT, we calculated GDP (Euro) as a function of TT (%) for the period 1993:3 to 1996:2 (the time window 2, see below).

## 4. Results

We present the results for time windows restricted to sections where tax policies lead the economy. We first show results for tax shocks, thereafter for the pair TT and GDP, and then for PT and UGE. Finally, we calculate the combined effects of tax reductions on GDP and UGE.

### 4.1. Tax Shocks

There were four shocks in personal taxes, seven shocks in corporate taxes and two shocks in social security taxes. Furthermore, there were shocks in tax control probability and UGE volatility. For the TT versus GDP pair, we used the sum of all tax shocks, as shown in Table 1.

### 4.2. Comparison of Time Series

The time series sets in Orsi et al. (2014) and in Chiarini et al. (2013) cover approximately the same time span, Figure 1c. However, whereas the Orsi series show an increasing linear trend, the Chiarini series show a decreasing linear trend. The detrended cyclic series show ordinary linear regression (OLR) characteristics: R = 0.28, $p = 0.009$, n = 88.

The ΔGDP and the UGE time series are shown in Figure 1d. The series shows OLR characteristics during the periods when PT leads UGE (red horizontal lines) as R = 0.50, $p = 0.003$, n = 33, and missing = 43.

### 4.3. GDP and Total Taxes

*LL relations*. Figure 3a shows the results for TT and GDP. We inverted the tax variable so that a peak in tax reduction (minus TT) is followed by a peak in the GDP (assuming, in agreement with our first hypothesis, that a reduction in TT will increase GDP). However, the LL relation holds for all segments of the time window, not only for the peaks. The line in bold shows time windows where -TT leads GDP. Visually, a peak in tax reduction will peak before a peak in GDP. The TT leads GDP 37% of the time.

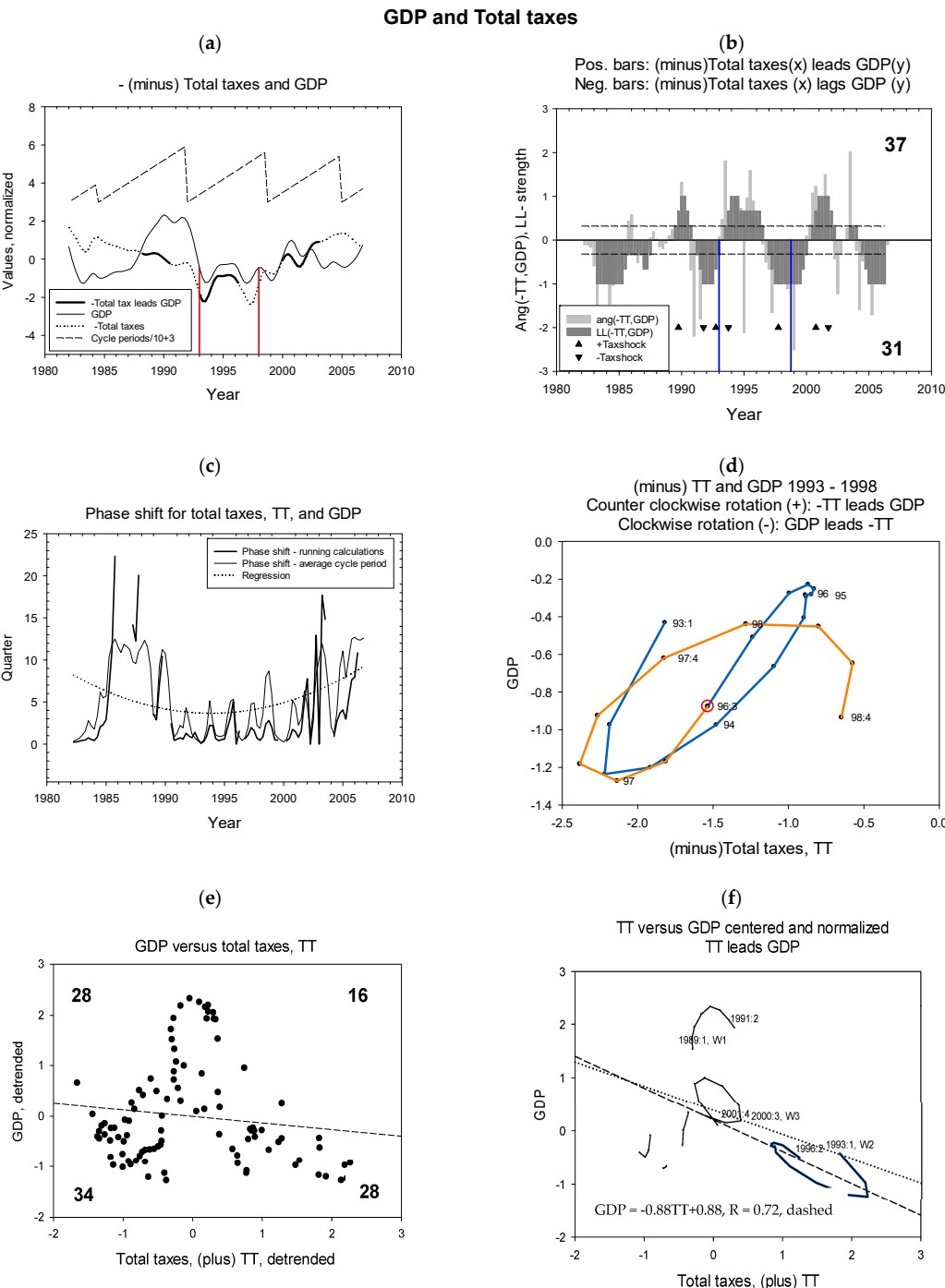

**Figure 3.** Lead–lag relations between total taxes, TT, and GDP. (**a**) Time series for reduction in TT (-TT) and GDP. Bold curve shows time windows where TT reduction leads GDP. Zigzag lines show cycle periods. Drop lines(the red lines) delineate the period 1993 to 1998. (**b**) LL relations between TT reduction and GDP. Light grey bars show LL relations with n = 3 and dark grey bars show LL relations with n = 9. Dashed lines show 95% confidence intervals. Triangles show positive and negative tax shocks. Drop lines(the blue lines) delineate the period 1993 to 1998. (**c**) Leading length (phase shift), calculation with running cycle period of 5 quarters and with average cycle period of 26 quarters. Leading length ranges between 3.7 and 8 quarters. (**d**) Phase plot for the period from 1993 to 1998. Note that rotational direction changes at about 1996:3 from counterclockwise rotation (+) to clockwise rotation (−). (**e**) GDP as a function of increasing total taxes, TT (all observations). Numbers show the number of states in each quadrant. (**f**) GDP as a function of increasing TT restricted to periods when TT reduction, -TT, leads GDP.

**Table 2.** Statistics for GDP and UGE.

| | Windows | Slope | $r^2$ | $p$ | n | Power | # Cycles |
|---|---|---|---|---|---|---|---|
| **GDP** [1] All observations | | −0.131 | 0.017 | 0.195 | 100 | 0.252 | |
| Total taxes lead GDP | | −0.833 | 0.486 | 0.001 | 100 | 0.999 | |
| 1 | 1988:2–1990:3 | 1.740 | 0.350 | 0.071 | 10 | 0.438 | 0.44 |
| 2 | 1991:1–1996:2 | −0.605 | 0.737 | 0.001 | 14 | 0.99 | 1.00 |
| 3 | 2000:1–2003:1 | 0.104 | 0.020 | 0.650 | 13 | 0.065 | 0.93 |
| | Weighted average | 0.413 | 0.369 | 0.241 | 12 | 0.498 | 0.79 |
| Average first 4Q | | −3.16 ± 2.671 | 0.38 ± 0.171 | 0.397 ± 0.142 | 4 | 0.111 ± 0.042 | - |
| **UGE** [2] All observations | | −0.11 | 0.013 | 0.271 | 99 | 0.195 | |
| Taxes lead UGE | | 0.31 | 0.075 | 0.123 | 33 | 0.337 | |
| 1 | 1986:2–1997:4 | −0.24 | 0.48 | 0.08 | 7 | 0.404 | 0.45 |
| 2 | 1995:3–1998:4 | −1.40 | 0.76 | <0.001 | 14 | 0.993 | 0.91 |
| | Weighted average | −1.01 | 0.66 | − | − | 0.80 | 0.76 |
| Average first 4Q | | 0.603 ± 0.994 | 0.349 ± 0.349 | 0.495 ± 0.303 | 4 | 0.115 ± 0.106 | - |

[1] **GDP**: Statistics for x = TT (total taxes) and y = GDP. (Both variables linearly detrended and normalized to unit standard deviation.) Windows 1 to 3 are time windows where taxes are leading variables for GDP. The statistics: slope or β- value, $r^2$ = explained variance, $p$ = probability, n = number of observations and power = the statistical sensitivity. # cycles = the number of cycles. Average first 4Q is the results for the four first quarters of each time window. [2] **UGE**: Statistics for x = PT (personal taxes) and y = UGE. Windows 1 to 2 are time windows where PT is leading UGE. Test statistics as for GDP.

Figure 3b shows the short-term LL relations (n = 3 quarters, light grey bars in the background) and the long-term LL relations (n = 9, dark grey bars). Figure 3b also shows, with triangles positioned upwards and downwards, when negative and positive tax shocks occur.

*Cycle periods and LL lengths.* The zigzag line in Figure 3a suggests cycle periods calculated by the cumulative angle method. There are three major common cycles for GDP and TT, each 10 to 14 quarters ≈ 2.5 to 3.5 years long. Figure 3c shows the two estimates for how long the leading time series is leading the lagging time series (GDP and TT shift in being leading and lagging).

The dashed line suggests that the LL time was about eight quarters at the beginning and at the end of the period, but about four quarters during the middle period. A phase plot for -TT and GDP during the five-year period from 1993 to 1998 is shown in Figure 3d. During this period, -TT shifts from leading GDP to lagging GDP, Figure 3b.

The trajectories in Figure 3d first rotate counterclockwise (+), showing that -TT leads GDP, then after 1996:3, the trajectories rotate clockwise (−), showing that GDP leads -TT. Neither of the (−) or (+) curves close.

*GDP as a function of TT.* Figure 3e shows a scatter plot for (plus) TT and GDP (100 quarters). A regression shows that increasing taxes decreases GDP. However, the result is not significant, $p$ = 0.195, and the power is << 0.800. The phase plot in Figure 3f shows two characteristics of the paired time series: (i) the β-coefficient shows how the y-value changes with the x-value and (ii) the rotational direction shows how the LL relation between the x- and the y-series change with time.

In Figure 3f, the scatter plot that only includes the observations where -TT leads GDP, the β-coefficient (the slope) is −0.88, the power is 0.999, and $p$ < 0.001. With higher taxes, above 32%, corresponding to 0 on the normalized TT-axis, the negative relation between taxes and GDP becomes pronounced. The numbers show the year and the quarter in the year of observation. No immediate effects of tax shocks appear in the response of GDP to tax changes. As a control, we also calculated the slope when GDP led the total tax series. The slope was then 0.00 and $p$ > 0.1. (Graph not shown; the observations correspond to those in Figure 3e minus those in Figure 3f).

*Shifting GDP backwards relative to TT.* If we, for the data set where -TT leads GDP, shift GDP two quarters backwards relative to TT, the explained variance increases from

$R^2 = 0.40$ to $R^2 = 0.60$ and the β-coefficient decreases from $-0.88$ to $-1.04$. The probability *p* is still $< 0.001$. (The tax series are inverted to describe tax reductions, but are positive in the x = tax, y = GDP graphs, and the time series are normalized to unit standard deviation; results not shown in the table.) With 10–14 quarter time windows, taxes will change, and GDP responds to the changes. The pair, TT and GDP, undergo cycles from $\frac{1}{2}$ cycle period to one full cycle period.

We examined the slope for GDP in terms of monetary units and TT restricted to the three windows where TT changes lead GDP. This slope is similar to the slope for time window W2; that is, the time window starting at the largest taxes ≈ 32% tax, Figure 3f. The equation is:

$$\text{GDP } (10^9 \text{ Euro}) = 0.185 - 2.32 \times \text{TT(\%)}, R = -0.775, p < 0.001, n = 14 \qquad (7)$$

The average GDP for the period 1982 to 2006 is EUR $216 \times 10^9$. A 1% decrease in TT would increase GDP by EUR $2.16 \times 10^9$, or 0.99% of the average GDP. (Note that the calculations are on the detrended series.)

Figure 1d shows a comparison of the first difference of GDP, $\Delta\text{GDP} = \text{GDP}_{t+4} - \text{GDP}_t$, compared to the detrended UGE. The horizontal lines show time windows where PT leads UGE. The rationale for the comparison is that with increasing GDP, which is $\Delta$GDP, more people may join the ordinary job market and leave the UGE. Table 2 also shows that the average response for the four first quarters is negative, although not significantly (the number of time steps, n, is only 4).

### 4.4. Underground Economy, UGE, and Personal Taxes

*LL relations.* Figure 4a shows the time series for PT and UGE. The bold sections of the PT series show the five time windows where PT leads UGE. The drop-down lines show boundaries for the time windows 1993 to 1998. In agreement with our assumption that an increase in taxes increases UGE, we here use tax increases as the x-variable in all four panels. PT leads UGE 21% of the time. Figure 4b shows the short-term LL relations (n = 3 quarters, light grey bars in the background) and the long-term LL relations (n = 9, dark grey bars).

*Cycle periods and LL lengths.* The zigzag line in Figure 4a suggests four common cycles for PT and UGE. The lead times for the leading series are from about three to about six quarters, Figure 4c. A phase plot of PT versus UGE during the period 1993 to 1998 is shown in Figure 4d. The trajectories rotate clockwise ($-$) from 1993 to 1994:3 and UGE leads PT, and then they rotate counterclockwise (+) to 1998 and PT leads UGE. The trajectories correspond to the LL relations in Figure 4b.

*UGE as a function of PT.* A regression shows that increasing personal taxes decreases UGE. However, the result is not significant, *p* = 0.271, and the power is << 0.800, as shown in Table 2 and Figure 4e. When we make a scatter plot only including the observations where personal taxes are leading UGE, increasing personal taxes increases UGE, and the result is significant, *p* = 0.03.

We made the regression with UGE (EUR $10^9$) and PT (%) for the time window W2 and obtained the following equation:

$$\text{UGE } (10^9 \text{ Euro}) = -1.4 + 0.887 \times \text{PT(\%)}, R = 0.87, p = < 0.001, n = 14, \qquad (8)$$

The average UGE for the period 1982 to 2006 is EUR $32.8 \times 10^9$. A 1% decrease in PT would decrease UGE with EUR $0.513 \times 10^9$; that is, 1.6% of the average UGE and 0.24% of the average GDP. (Note that the series are detrended.) However, when we examine the graph in Figure 4f, there are five separate datasets where PT leads UGE. Only two of the sets have lengths larger than three quarters. For the long series we get a slope of 0.89.

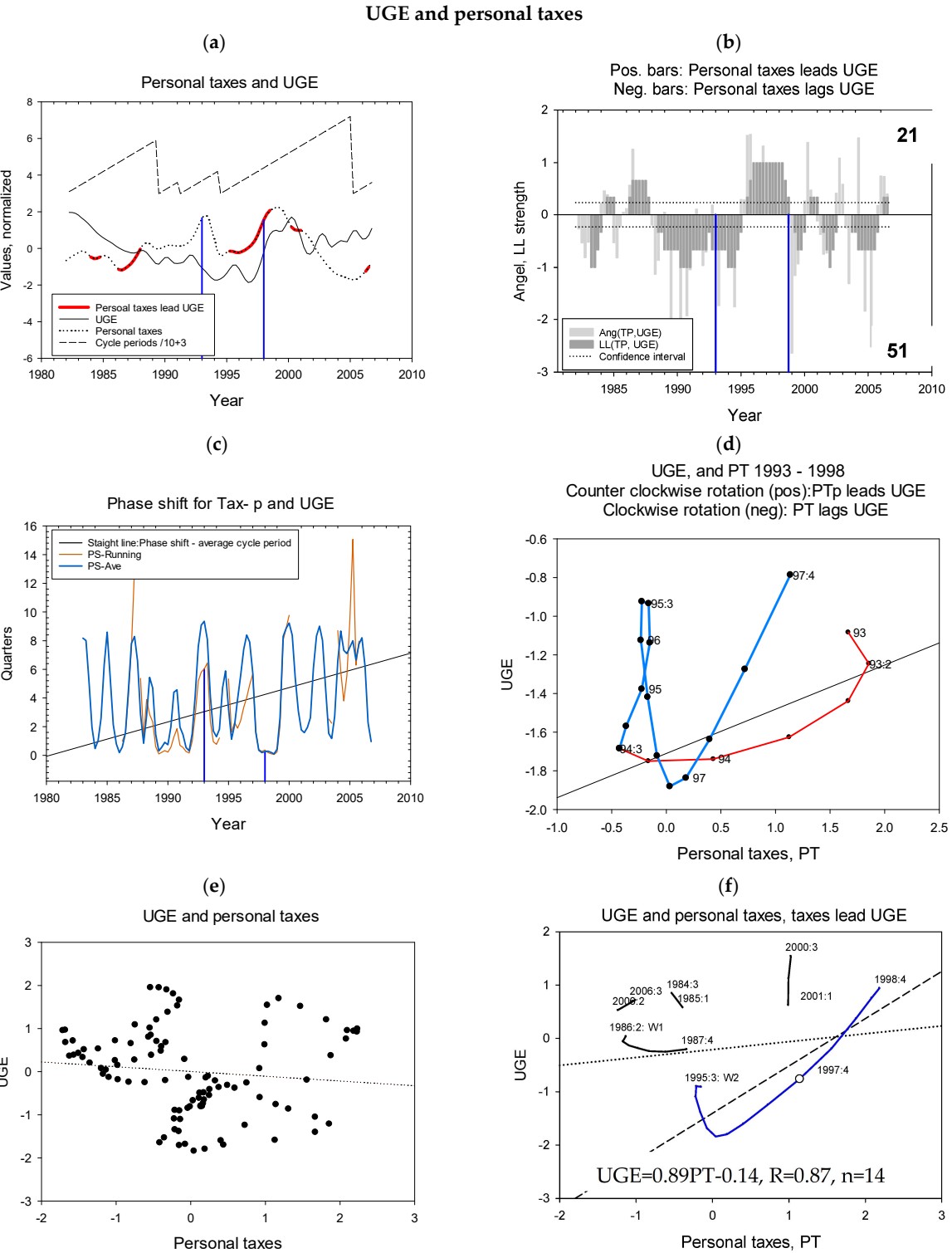

**Figure 4.** Leading and lagging relations between personal taxes, PT, and the underground economy, UGE. (**a**) Time series for personal taxes and UGE. Bold curve shows time windows where PT leads UGE. The zigzag line shows cycle periods. Colored droplines here and in panels (**b,c**) show sections of personal taxes and UGE series that is depicted in the phase plot in panel (**d**). (**b**) LL relations between PT and GDP. Light grey bars show LL relations with n = 3 and dark grey bars show LL relations with n = 9. Dashed lines show 95% confidence intervals. (**c**) Lead times (phase shifts) for PT versus UGE. (**d**) Phase plot for PT and UGE during the period between 1993 and 1998 Q4. (**e**) UGE as a function of increasing PT (all observations). (**f**) UGE as a function of increasing PT restricted to periods when PT leads UGE.

We also calculated UGE as a function of ΔGDP when PT was leading UGE. The equation became:

$$\text{UGE } (10^9 \text{ Euro}) \; 24.8 + 9.42 \times \Delta\text{GDP (Normalized to unit st.dev.), R} = 0.5, \tag{9}$$
$$p < 0.003, \text{n} = 33, \text{missing} = 43$$

*Shifting UGE backwards relative to PT*. If we shift UGE 2 quarters backwards (the full data set in Figure 4e), corresponding to the lag in the effect on tax reductions on UGE, the explained variance increases from $R^2 = 0.09$ to $R^2 = 0.18$ (statistics not shown in the table), the β-coefficient increases from 0.27 to 0.38, and the *p*-values decrease from 0.029 to 0.004. With time series 5–14 quarters long, taxes change, and UGE responds to the changes. For the pairs PT and UGE, the pairs undergo cycles from 1/3 cycle to one full cycle. Only the shock in PT in 1997:4 (+3.8) affects PT and UGE in a time window where PT → UGE. Table 2 also shows that the average response for the four first quarters is positive, although not significantly (n = 4).

### 4.5. Self-Financing Rate

The loss in tax revenues by decreasing the tax rate from 32% to 31% is EUR $0.76 \times 10^9$. First, we found in Section 4.3 that a 1% decrease in TT would increase GDP by 0.99%. The tax revenues from this increase, 31% of the increased GDP, will compensate for lost tax revenues by $0.99\% \times 0.31 = 0.31\%$. Second, we found that a reduction in personal taxes of 1% decreases UGE with 1.6%, corresponding to 0.24% of GDP. Together, the two contributions give an increase in GDP of 0.31% + 0.24% = 0.55%. Thus, reducing total taxes on GDP and personal taxes on UGE gives added revenues. The compensation factor for reducing taxes with 1% is thus 0.55%, and both factors contribute about equally.

## 5. Discussion

The economy acts on different time scales. Economic trends over multidecadal scales are more likely caused by technical or social innovations (Bloom et al. 2013; Hamilton 2018), and improvements in infrastructures are dependent on long-term tax incomes. In contrast, shorter decadal events could be caused by tax policy changes. In this section, we discuss relations on interannual and decadal time scales when taxes lead the economies.

### 5.1. Detrending, Smoothing, and Normalizing

To identify business cycle length variability in the series, we linearly detrended the data. For the period discussed here, we believe that a linear detrending is sufficient. However, strongly HP-smoothing the time series and using the residuals from the smoothed series as representative of decadal variability could have been an option (e.g., Chiarini et al. 2013). The relevance of business cycle periods of about five years for abatement measures to reduce UGE is also proposed by Goel et al. (2019, p. 102).

### 5.2. Comparing UGE Estimates

Estimating the size and variability of tax evasion (the underground economy, the shadow economy, or the dark economy) is difficult (Dell'Anno and Davidescu 2019). We have two independent estimates for tax evasion for the period 1982 to 2006. The two estimates express different aspects of UGE and tax evasion, but comparing them may still bring support for the size and variability of UGE. The residuals of both series showed a reasonable overall similarity (R = 0.28, *p* < 0.009). However, Chiarini et al. (2013) defined their trend as a HP-filter approximation to their tax evasion time series, and it is not linear. We cannot explain why we found a positive linear trend for the Orsi et al. (2014) estimate and a negative trend for the Chiarini et al. (2013) estimate based on VAT measurements. However, Dell'Anno and Davidescu (2019, p. 136) show that tax evasion and the shadow economy were counter-cyclic in Romania during the period 2000 to 2017. Chiarini et al. (2013, p. 279) compared their tax evasion series to tax events and concluded that patterns observed in the series can be explained by "structural innovations" related to the system.

These arguments extend to the UGE series obtained by Orsi et al. (2014), since the two series show good similarity on a business cycle scale.

### 5.3. GDP Results

We posited three hypotheses with respect to taxation, GDP, and UGE. The first, **H1**, was that reducing tax rates would increase GDP. We show that by decreasing the overall tax burden, GDP will increase. This effect is most pronounced during the first four quarters after the tax policy becomes a leading variable to GDP. Chiarini et al. (2022) suggest, with data from the US, that corporate tax evasion (which is part of our TT) may change the allocation of assets between consumption and productivity uses and thus also affect the business cycle. However, Ngelo et al. (2022) found that Indonesian firms that have a tax avoidance practice utilize the extra cash to invest in value-enhancing projects.

The cyclic components of TT and GDP have common cycle periods of about 20 quarters. Lead times between the two series range between 1 and 10 quarters, with averages of 3.1 and 4.6 quarters depending on the calculation method.

Since the effects of tax changes take time, the time series for GDP were shifted backwards in time relative to -TT. Shifting the series relative to each other caused the β-coefficient to decrease from $-0.88$ to $-1.04$, and the explained variance to increase. However, using one overall shift for the whole series is only an approximation, since the lead time between -TT and GDP change in size and in direction. The lead time using cross-correlation techniques resulted in a lead time of three quarters. This is a little less than the average lead time for the whole series. One would assume that the effect of reducing taxes depends upon the tax level at the time of tax changes. Our result suggests that the effect of reducing taxes has been most effective at tax levels above 32% in Italy during the period 1982 to 2006 (see Figure 3f). At lower tax levels, taxes and GDP start cycling with cycle periods of about 2–3 years (8–12 quarters), probably because low tax levels are not sustainable either economically or politically.

### 5.4. Underground Economy (UGE) Results

*LL relations.* Our second hypothesis, **H2**, states that changing the tax level influences the size of tax evasion, or UGE, in Italy. Here, we found that tax reductions would decrease UGE significantly. However, since the effects of tax changes take time, the time series for UGE were shifted backwards in time relative to PT. Shifting the two series (restricted to the windows where PT leads UGE) relative to each other caused the β-coefficient to increase from 0.27 to 0.38. The explained variance, $R^2$, increased from 0.09 to 0.18 and the probability, p, decreased from 0.03 to 0.004. We also examined tax evasion control probability but found conflicting results for short- and long-term effects. One reason may be that tax amnesties in Italy may be ineffective. Alstadsaeter et al. (2022) found that an amnesty, combined with increased probability of detecting undeclared (offshore) accounts, increased tax revenues in Norway. Mara (2021, p. 319) found that tax-rate increases on labor increased UGE significantly ($p < 0.05$), except in Mediterranean countries where the result was insignificant. The study included annual data 1995–2017 for 28 European union countries.

We found that UGE increased with positive changes in GDP. The result contrasts with the finding that undeclared work is more prevalent in countries with lower GDP (Mara 2021). Goel et al. (2019, p.101) found that GDP (and inflation) did not significantly affect UGE. One reason for the effects of changes in UGE may be that with growth in GDP more opportunities become available for work in sectors that traditionally employ workers in the underground economy, such as the personal service sector, which is 27% of GDP in Europe (Williams and Kayaoglu 2020, p. 85).

In the literature, there is conflicting evidence whether increasing tax rates increase tax evasion or not, and a summary from 1980 until 1995 is given in Ali et al. (2001). Empirical studies normally address long time series. Ali et al. (2001) used annual US data from 1980 to 1995; Cebula and Feige (2012) used US data from 1960 to 2008, but examined different time windows, the smallest time window being 28 years. These authors found a significant

positive correlation between compliance and tax rate and between compliance and penalty rate, although for some results these variables had to interact with screening variables, such as actual income (Ali et al. 2001; Cebula and Feige 2012). Overall, the literature studies are different in their assumptions, and therefore direct comparison with our results is not straightforward. A series of other tax evasion abatement measures can also be envisaged (Luttmer and Singhal 2014; Mascagni 2018). Lisi and Pugno (2015) conclude in a model study that closer monitoring of tax evasion attempts and lower taxation reduces the underground economy. However, our examination of tax evasion control probabilities in Italy was not conclusive.

*5.5. Effect on Tax Revenues from Changes in GDP and UGE*

Our third hypothesis, **H3**, that the loss in TR from decreased taxes would be compensated by increased taxes from enhanced GDP and decrease in the underground economy, UGE, was not supported. When tax levels were relatively high, at >32%, and GDP low, the compensation factor was only 0.55 to 1.00. The GDP contributed most (GDP: 56%, UGE 44%). Since GDP and unemployment are often counter-cyclic, e.g., Okun's law, the relevant information may not be low GDP, but that there is a high unemployment rate. Our results are based on observations and thus dependent on the conditions in the Italian economy in the period 1982–2006. Studies of the Laffer curve for Italy suggest that the maximum labor tax rate is between 41% and 62%; Ferreira-Lopes et al. (2018) and Trabandt and Uhlig (2011), respectively. A similar Laffer curve value, around 60%, was found by Busato and Chiarini (2013, p. 620).

*5.6. Tax Shocks and Tax Policies*

It turned out to be difficult to relate tax shocks to changes in GDP or UGE. The reason is probably that tax shocks may have been announced or anticipated, or that several minor changes in tax levels have been more important than the shocks. In addition, we do not know the effects of tax evasion amnesties during the period. It is difficult to find proxies for all information about future shocks that policy makers and corporation may have (Romer and Romer 2010). Hayo and Mierzwa (2022, p. 5) found that drafting tax bills influenced GDP in the USA, the United Kingdom, and Germany, data from 1977 to 2018. Chiarini et al. (2013, p. 279) argue that there are causal relations between tax policy decisions and short-term effects on the time series for tax evasion. These authors also found breaks in the economy in 1983:4 and in 1998:1. The inclusion of UGE in our evaluation of tax change effects was supported by Annicchiarico and Cesaroni (2018) who found that neglecting UGE may lead to an underestimate of the effects of tax reforms, thus emphasizing the effect of tax reductions also on UGE.

We found three GDP time windows and two UGE time windows longer than three quarters where taxes were leading GDP or UGE during the period 1982 to 2006. The first half of the 1990s showed high political instability, whereas the period 1996 to 2000 gave a more stable framework for tax collection. However, in 1997:4 there were tax shocks in all tax variables. Personal and corporate taxes were increased, whereas the social security tax and control probability were reduced. The events in 1997:4 may be due to the reorganization of the fiscal authority that started in 1997. The tax shocks in 1997:4 were included in the W2 time series for the UGE = f (PT) regression. For some tax shocks that were followed by a leading role for tax rates, there were political events that could help explain changes in tax policies, as in the year 1997. However, TT and PT were simultaneous leading variables for GDP and UGE only 10% of the time.

Several authors suggest that the effect of rapid changes in tax levels have typical time horizons of 4 to 6 years, e.g., Mountford and Uhlig (2009) and Romer and Romer (2010). We found that the time horizon varied between 2 and 6 years with an average of thirteen quarters. Thus, the LL method identifies potential causal relations between tax policy changes and the economies of similar lengths as found in the literature.

### 5.7. Robustness

Our calculation of self-financing of tax reductions has several caveats. First, taxes are complex constructs and using two tax levels (TT and PT) may oversimplify the real tax structure. Tax evasion by corporations may alter the wealth transfer between households and corporations, implying a reallocation of assets between consumption and productivity uses (Chiarini et al. 2022). Personal taxes in Italy are progressive and will influence groups differently. The components of the total taxes and their enforcement, Sepulveda (2023), act on different parts of the economy and determine how tax revenues are used, Alinaghi and Reed (2021, p. 14).

There may be several variables that have significant LL relations to each other, and for n series there will be m = n(n − 1)/2 pairs that can be compared. Seip and Grøn (2018) show how several LL relations can be interpreted by identifying time windows where LL relations are persistent. We assume that a persistent LL relation between two variables strengthens a cause–effect relation. However, LL relations may be due to third factors that influences the two variables, but one later than another, so that there appears to exist a LL relation. Thus, it is necessary that auxiliary, e.g., mechanistic, information exists that support a cause effect. Furthermore, it may be difficult to establish a "ground truth" for LL relations between real time series. An example where this was possible is a study on economic forecasting in Germany. The leading forecasting series was leading 78% of the time, which is in compliance with characteristics of leading series in economy, Seip et al. (2019). The results are based on several assumptions; for example, that all taxes recovered are used to increase GDP. However, some of the taxes may be used for other purposes, such as paying down government debt.

### 5.8. Policy Implications

At low tax rates, both the pair TT-GDP and the pair PT-UGE start to cycle. However, with high total taxes, around > 32%, low GDP, or high unemployment; reducing total and personal taxes give a fiscal multiplier of about 0.55 to 1.00. Thus, the benefit of a tax reduction policy would not be sufficient to compensate for lost tax revenues in Italy during the period 1982 to 2006. Although the results were obtained for a period two decades ago, the economy and the tax regimes in Italy have probably not changed to an extent that would invalidate the conclusion, e.g., Astarita et al. (2016) on recent tax reforms in Italy. Adding measures that reduce tax evasion may increase the multiplier (Moro-Egido and Solano-García 2020). Thus, with the high taxes that currently (2023) prevail in Italy, our study suggests that taxes could be reduced in concert with other tax policy measures, such as increased tax enforcement. This conclusion is in line with results from Acocella et al. (2020), who advise that tax compliance should be strengthened. Generic advice to increase tax compliance is also given by Moro-Egido and Solano-García (2020), Lisi and Pugno (2015), and Alstadsaeter et al. (2022), although Italy is not explicitly included in their studies on tax compliance.

### 5.9. Further Work

We have used two aggregate measures of taxes, but different taxes will act differently on the economy. Thus, we believe that a more detailed treatment of taxes, and a closer examination of how they affect the economy, would be beneficial. Further, some taxes may have two or more objectives, e.g., improving infrastructure or changing inequality levels (long-term goals) and boosting short-term GDP. How multiple goals should be balanced could be included in further work.

On the technical side, since several factors affect GDP, such as the central bank's short-term rent (CBR) as well as tax level, it would be interesting to see if only those periods in the economy where tax changes lead GDP, but CBR does not, would change the result.

## 6. Conclusions

There are several ways in which tax changes can influence the economy, and both positive and negative effects may be envisaged. Our contribution consists of applying a novel test to the effects of tax changes based on empirical data. We have achieved this by (i) removing multidecadal trends, (ii) restricting the study to time windows where tax changes are leading the economies and thus have a high probability of affecting the economy, (iii) examining both GDP and UGE, and (iv) including the finding that tax changes take time to affect GDP and UGE by shifting GDP and UGE backwards in time before regressions are applied.

We found that there are complete cycles between tax changes and GDP changes; that is, tax changes beget tax changes. With high total taxes > 32%, decreasing taxes increased GDP. The results for PT and UGE showed that increased personal taxes increased UGE. Tax policies were leading GDP and UGE for two to six quarters corresponding with literature values for the effects of tax policy variables. Our empirical findings provide useful insights into the effectiveness of policies aimed at reducing the tax burden to obtain incentives for economic growth and reduced UGE.

**Author Contributions:** Conceptualization, R.O. and K.L.S.; methodology, K.L.S.; software, K.L.S.; validation., R.O. and K.L.S.; formal analysis, R.O. and K.L.S.; investigation, R.O. and K.L.S.; resources, R.O.; data curation, R.O.; writing—original draft preparation, R.O. and K.L.S.; writing—review and editing, R.O. and K.L.S.; visualization, K.L.S.; supervision R.O.; project administration, R.O. and K.L.S.; funding acquisition, K.L.S. All authors have read and agreed to the published version of the manuscript.

**Funding:** This research was funded by Oslo Metropolitan University.

**Informed Consent Statement:** Not applicable.

**Data Availability Statement:** All data and all essential calculations are available from the second author, K.L.S.

**Conflicts of Interest:** The authors declare no conflict of interest.

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
