# Peer review of "Do Increased Tax Base and Reductions in the Underground Economy Compensate for Lost Tax Revenue Following a Tax Reduction Policy? Evidence from Italy 1982 to 2006"

_economies, doi:10.3390/economies11070177_

Round 1
Reviewer 1 Report
The goal of the submitted paper has yet to be explicitly stated. But one can deduce that the paper aims to evaluate the relationship between GDP growth, the underground economy changes, and total and personal taxation changes. The article needs to address many serious problems. It needs to be clarified the purpose and goal of the paper and its contribution to the existing literature. The presented and used “novel high-resolution lead-lag technique” is only very weakly motivated, and there is a complete lack of explanation of its suitability compared to existing standard methods and techniques. The data used are very weakly described. The figures, tables and equations are very unclear and poorly labelled. The presentation and discussion of the empirical results are very technical and purely descriptive. The conclusions are mostly speculative.
The article gives me the impression that it presents one particular data-handling technique in great detail and applies it marginally to the economic problem, which could be somewhat related to the paper’s title. This method has been used repeatedly in other articles (with a recurring co-author who is probably one of the authors of this paper as well). So, the only difference is in the data it has been working with. In the case of the presented article, however, the substantive (economic) nature of the topic under investigation is heavily suppressed in favour of the pure application of a single technique.
My overall evaluation is that the submitted paper needs to be better written and more transparent. It requires an explanation for motivation and a detailed contribution to the existing literature. My major comments may be summarised as follows.
1. The goal and motivation of the paper are missing. There is no link to the existing literature. It is hard to find a contribution to the literature. The statement that “The present study can be distinguished from other studies in that all series are linearly detrended” does not seem as relevant. Working with detrended data is a standard approach in many applications.
2. What are the novelty and the main benefits of the used LL technique? Does it have some empirical or theoretical advantages over the other approaches? (SVAR models, VAR models, cross-correlation approach, Markov-Switching models, non-linear time-series models)
3. Data and the data sources are poorly described, and one cannot replicate the results (or test their plausibility). Moreover, the data on the size of the underground economy are model-based estimates (such as Orsi et al., 2014) and entirely related to the other underlying model variables (GDP growth/gap, total and personal tax rate). F. Schneider presents estimates of the shadow economy in developing countries as well. These data may help to improve the plausibility and robustness of the results.
4. The presented graphs in Figure 1 are unclear. The labels and the overall explanation must be improved. What is the meaning of UGH+3, Total tak+6 etc.? Does it mean the lag or lead of these series? The meaning and purpose of the trend lines in Figure 1c are unclear.
5. The tax shocks were identified as outliers of Normal distribution. Why must one believe that the tax changes should follow the Normal distribution? Moreover, the tax rates seem to be used as a share of GDP (as implied by rows 80 and 81). The changes will be thus caused by many other economic factors that influence GDP growth. I have, therefore, a significant concern about the validity of the obtained shock variables.
6. The very similar description of the LL method may be found in other articles (including the same illustration exercise based on random data).
7. The regression equations are confusing (especially the labelling in parentheses). Due to the endogeneity of the regressors, there will be a severe bias in parameter estimates (and the implied results as well)
8. The authors sometimes use computation based on averages of level (non-stationary variables). Does it is appropriate for non-stationary variables?
9. The presented results of section 4 are a little confusing. It is primarily a technicist description of the tables and figures without evaluating their relevance and importance for the investigated topic.
10. The most crucial part of section 5 is related to the robustness of the results. The problem of omitted variables is critical. Without testing it, the results and the corresponding implications have little credibility.
11. The policy implications are somewhat speculative. The extrapolation of the results to the next two decades is highly questionable.
Reviewer 2 Report
The presented article is very well written and structured. The study offers new evidence and contributes to tax scientific field. The methodology chapter is very clear (maybe I would suggest to the author(s) to consider to change the name of the chapter to Methodology). I fully recommend to publish the paper.
For the authors I have only few minor recommendations to be considered:
- the term underground economy is explained in more details in lines 552 and 553 (subchapter 5.2). I suggest to place these synonyms somewhere at the beginning of the paper.
- to check the part of the sentence in lines 714 and 715 ... tax changes beget tax changes ... is it correct? not GDP changes?
- In References, for some sources (Journals) are missing the information about the volume and number of pages - I suggest to add the missing information and to unify the way of presentation of the sources.
Reviewer 3 Report
In the paper, an interesting topic is addressed, considering the correlation between the level of taxation, the collected tax revenues, tax evasion and the underground economy. The authors may consider the following suggestions:
- rewriting the abstract so that the objectives of the research, the research methods applied and the results obtained are clearly highlighted;
- the introduction can be supplemented with clearer explanations regarding the topic addressed and more results obtained by other authors, considering the fact that a distinct part is not allocated for the analysis of the specialized literature;
- the clear explanation of the variables used (for example, what are the taxes included in the PT category? Are they those that affect the income and wealth of natural or legal persons? From the authors' explanations it appears that they also refer to indirect taxes, such as VAT; or the indirect does not fall into the category of personal taxes);
- to explain all the variables in the used equations;
- subsection 3.4 is insufficiently explained, in correlation with its title (self-financing is assimilated, as a rule, with the part of the profit used for development; or, the authors probably want to report the level of tax revenues collected in the context of reducing the tax rate by 1% ); ditto for 4.5 where the self-financing rate is analyzed (what is the self-financing rate? it would be much clearer for any reader if it were clearly highlighted as a method of calculation); in addition, the work is circumscribed to an analysis at the level of fiscal revenues collected by the state, without clearly explaining why it is considered that all the savings resulting from the reduction of the level of taxation are allocated to development, with a direct impact on the GDP;
- in the Conclusions section, the limits of the research and future research directions can be specified.
Round 2
Reviewer 1 Report
I thank the authors for their detailed explanatory comments on the revisions made. As well as for providing the underlying data calculations that helped clarify some issues. However, for better clarity and flow of the text, I would consider whether it is necessary to include all technical details and illustrative examples ((including appendices) that are not directly related to the topic of the paper and can be found in other publications.
Author Response
Dear referee
Thank you for your considerable efforts in reviewing our paper. We agree with you that the appendix and the supporting material can be deleted.
We acknowledge that there are technical details and illustrative examples in the main text, and it would have been nice if we could have made the paper shorter. However, to clearly see the advantage of limiting the tax, GDP and UGE series to time windows where taxes lead GDP and UGE we think the examples in the main text are useful.